# Neuromyelitis Optica Spectrum Disorder: From Basic Research to Clinical Perspectives

**DOI:** 10.3390/ijms23147908

**Published:** 2022-07-18

**Authors:** Tzu-Lun Huang, Jia-Kang Wang, Pei-Yao Chang, Yung-Ray Hsu, Cheng-Hung Lin, Kung-Hung Lin, Rong-Kung Tsai

**Affiliations:** 1Department of Ophthalmology, Far Eastern Memorial Hospital, Banqiao Dist., New Taipei City 220, Taiwan; jiakangw2158@gmail.com (J.-K.W.); peiyao@seed.net.tw (P.-Y.C.); scherzoray@gmail.com (Y.-R.H.); 2Department of Electrical Engineering, Yuan Ze University, Chung-Li, Taoyuan 320, Taiwan; chlin@saturn.yzu.edu.tw; 3Department of Medicine, National Yang-Ming University, Taipei City 112, Taiwan; 4Department of Medicine, National Taiwan University, Taipei City 106, Taiwan; 5Department of Healthcare Administration and Department of Nursing, Oriental Institute of Technology, New Taipei City 220, Taiwan; 6Biomedical Engineering Research Center, Yuan Ze University, Taoyuan 320, Taiwan; 7Department of Neurology, Taiwan Adventist Hospital, Taipei City 105, Taiwan; kksao.lin@gmail.com; 8Institute of Eye Research, Hualien Tzu Chi Hospital, Buddhist Tzu Chi Medical Foundation, Tzu Chi University, 707 Sec. 3 Chung-Yung Road, Hualien 970, Taiwan; 9Institute of Medical Sciences, Tzu Chi University, Hualien 970, Taiwan

**Keywords:** neuromyelitis optica spectrum disease, aquaporin-4, myelin oligodendrocyte glycoprotein, ocular coherence tomography, complement, microcystic macular degeneration, Müller cell, astrocyte, oligodendrocyte, microglia

## Abstract

Neuromyelitis optica spectrum disorder (NMOSD) is an inflammatory disease of the central nervous system characterized by relapses and autoimmunity caused by antibodies against the astrocyte water channel protein aquaporin-4. Over the past decade, there have been significant advances in the biologic knowledge of NMOSD, which resulted in the IDENTIFICATION of variable disease phenotypes, biomarkers, and complex inflammatory cascades involved in disease pathogenesis. Ongoing clinical trials are looking at new treatments targeting NMOSD relapses. This review aims to provide an update on recent studies regarding issues related to NMOSD, including the pathophysiology of the disease, the potential use of serum and cerebrospinal fluid cytokines as disease biomarkers, the clinical utilization of ocular coherence tomography, and the comparison of different animal models of NMOSD.

## 1. Introduction

Neuromyelitis optica spectrum disorder (NMOSD) is a common cause of optic neuritis (ON) in Taiwan. In 2015, the prevalence of NMOSD was 1.47/100,000, and the age-standardized annual incidence rate was 0.61/100,000 person-years [1]. The reported prevalence of NMOSD in different racial groups is approximately 1/100,000 in White individuals, 3.5/100,000 in Asians, and 10/100,000 in Black individuals [2]. The differential diagnosis of NMOSD and multiple sclerosis (MS) was challenging until the discovery of neuromyelitis optica (NMO) autoantibodies by Lennon et al. [3,4]. In most cases, NMOSD is caused by pathogenic NMO immunoglobulin G (IgG) autoantibodies that bind to the aquaporin-4 (AQP4) target antigen, a water channel expressed on the end-feet membranes of astrocytes along the blood–brain barrier (BBB) and in Müller cells distributed on the fovea centralis in the retina [4,5,6,7,8,9]. The pathology most often occurs in the periventricular zone, involving astrocyte plasma membrane domains facing the pia and vessels, whereas the least-affected site in the central nervous system (CNS) is the area postrema in the dorsal medulla [10,11]. 

Currently, the clinical diagnosis of NMOSD is mainly based on the detection of serum NMO-IgG (AQP4-IgG) antibodies and the presence of core symptoms included in the diagnostic criteria developed by the International Panel for NMO Diagnosis in 2015 (Table 1) [10,12,13]. The revised criteria that replaced the previous 2006 criteria for NMO diagnosis resulted in a significant increase in the diagnostic sensitivity of NMOSD by 76% (62% in the AQP4-IgG-positive group and 14% in the seronegative group) [14]. For AQP4-IgG-positive patients, at least one of six sites within the CNS, including the spinal cord, optic nerves, area postrema, diencephalon, brainstem, and cerebrum, must be attacked. In seronegative patients, at least two core sites have to be affected and additional magnetic resonance imaging (MRI) criteria fulfilled [13]. The rate of seropositivity for myelin oligodendrocyte glycoprotein (MOG-IgG) antibodies in AQP4-IgG-seronegative patients with NMOSD was reported to reach up to 41.6% [15].

From the perspective of clinical application, biological biomarkers may be important for predicting the future risk of relapse and disease prognosis [10,16]. AQP4-IgG antibody titers seem to be linked to clinical presentation and immune response, with higher titers associated with worse visual function and more extensive cerebral involvement on MRI [16]. On the other hand, AQP4-IgG antibodies might represent a byproduct resulting from complex immunoinflammatory processes in NMOSD, with no significant variations in antibody titers between different disease stages [17]. Beyond autoantibodies, the clinical presentation and demographic features may be more reliable in terms of prognosis prediction [18]. Age was reported to be predictive of the involvement site, and ON seems to be the most common inflammatory lesion in NMOSD patients younger than 30 years [19].

AQP4-IgG-seropositive NMOSD indicates the entity of astrocytopathy, and MOG-IgG is a protein expressed by oligodendrocytes on the most superficial surface of myelin sheaths, which results in oligodendropathy [20,21,22]. MOG antibody-associated disease (MOGAD) exhibits different clinical features from those of AQP4-IgG-seropositive NMOSD [23,24,25]. The phenotype difference between AQP4-IgG- and MOG-IgG-positive ON can be assessed by the length of the optic nerve involvement and preferable involvement site on MRI, the morphology of the optic disc, laterality, and the pattern of the ganglion cell–inner plexiform layer (GC-IPL) on optical coherence tomography (OCT) [24,26]. MRI image characteristics add evidence to the differential diagnosis of ON. AQP4-ON preferentially presents with a longer, more unilateral, more posterior portion of the optic nerve with T1 gadolinium enhancement [13,27,28]. In contrast, MOG-ON usually presents with a longer, more bilateral and more anterior portion of the optic nerve accompanied by intraorbital optic nerve swelling and perineural T1 gadolinium enhancement [29].

Sex difference is low due to the higher proportion of males in the MOGAD group compared with that in the AQP4-IgG group. For MOGAD, 41% (7/17 case) to 44% (4/9 case) of female cases were noted [24,30]. MOG-ON may present at multiple ages and shows no sex bias, but the female/male ratio is 7.2:1.0 in the AQP4-ON group [26,29,31]. Moreover, only 9% of MOG-IgG-positive cases have a concurrent autoimmune disorder, and 80% of MOGAD cases will relapse but have a better clinical prognosis [32,33].

Although most neuronal damage occurs during the first episode, the treatment of a relapse episode in patients with NMOSD is essential for preserving as much of the neuron reservoir as possible. Most patients with NMOSD achieve good functional improvement after corticosteroid treatment and add-on plasmapheresis in the acute stage. However, clinical relapse occurs in most cases, resulting in cumulative neurologic damage. The new strategies may provide additional options for patients who are refractory to current maintenance therapies including treatments interfering with eosinophilic function, monoclonal antibodies that target neutrophil elastase, complement activation, interleukin IL-6 receptor (IL-6R) signaling, and plasma cells producing AQP4, and MOG-IgG antibodies [34,35,36,37,38,39,40]. Recently, three monoclonal antibody therapies approved by the Food and Drug Administration for the treatment of NMOSD demonstrated safety and efficacy in reducing the risk of relapse during remission; these are eculizumab (inhibitor of complement protein C5), inebilizumab (humanized monoclonal antibody against CD19 B cell protein), and satralizumab (humanized recombinant monoclonal antibody targeting IL-6R).

## 2. The Pathogenesis of NMOSD

AQP4 contributes to the stabilization of extracellular osmolality during neuronal activity. Moreover, it maintains glutamate homeostasis and energy balance as well as buffers the metabolic load in the BBB [5,41]. The pathological features of NMOSD include activated complement with extensive vasculocentric immune complex deposition, the loss of AQP4 expression in astrocytes, neutrophil/macrophage/microglial infiltration and eosinophil degranulation, myelin loss, and thickened hyalinization blood [4,5,42,43,44,45].The two major AQP4 isoforms, M1 and M23, exhibit locational and maturational differences in the ratio of M1 to M23 proteins along the astrocytic membrane, which possibly determines the pathogenicity and a different anatomical distribution in the CNS and at different stages of CNS maturation in pediatric and adult patients [46,47,48,49]. The proportion of the largest AQP4 aggregate is the highest in the optic nerve followed by the spinal cord; this is relevant to why NMO selectively targets the CNS tissue and spares non-CNS AQP4-expressing tissues [50]. The M1 protein is completely internalized, but M23 resists internalization and activates the complement more efficiently than M1 when bound by the antigen [46,51]. The relative components of AQP4 isoforms are tissue-specific, with an approximate 3:1 ratio of AQP4-M23 to AQP4-M1 in rat brain [52]. Formation of supramolecular structures, called orthogonal arrays of particles (OAPs), by AQP4 is essential in NMOSD pathogenesis and enhances complement-dependent cytotoxicity (CDC) by the pathogenic AQP4-IgG [53]. It remains unclear if the OAP composition varies in pediatric and adult patients or whether OAP differences may cause different phenotypes [54].

Müller cells and neuronal axons are the main targets in an experimental model of NMO [55]. After intravitreal injection of AQP4-IgG antibodies, complement activation and immunoglobulin deposition was found in Müller cells and caused a retinal pathology [56]. AQP4 is also coexpressed with the Kir4.1 potassium channel subunit in cells, and the electrogenic bicarbonate transporter contributes to changes in the extracellular space, involved in buffering K+ [8]. In clinical practice, Müller cell dysfunction was shown to significantly reduce the b-wave amplitude in the scotopic electroretinogram of AQP4-IgG-positive patients compared with normal controls [57]. However, the results were inconclusive regarding a relationship between the b-wave amplitude and the volume of the outer retinal segment on OCT as well as disease severity, assessed on the basis of the Expanded Disability Status Scale (EDSS) or visual acuity [57].

## 3. Cytotoxicity Pathway

Complement-dependent cytotoxicity (CDC), complement-dependent cellular cytotoxicity (CDCC), and antibody-dependent cellular cytotoxicity (ADCC) are responsible for astrocyte injury in NMOSD [58,59]. ADCC seems to play a main role in facilitating macrophage and natural killer (NK) cell activation after binding to the CH3 region of IgG antibodies via the effector cells’ Fc receptors in the outer zone of developing lesions (penumbra) [59,60,61]. In CDC, antibody binding to a target antigen triggers the classic complement pathway and results in the formation of the membrane attack complex (MAC). In CDCC, another protein, C3b, is expressed during the complement cascade activation and interacts with NK cells and macrophages to facilitate cell lysis [62].

Peripheral autoimmune dysregulation starts after the modulation of peripheral T cells. Pathogenic autoreactive T cells (Th17 cells) and IL-6 disrupt BBB tight junctions, resulting in CNS inflammation due to the effect of numerous chemokines and cytokines (Figure 1) [63,64,65]. IL-6 is a proinflammatory cytokine that amplifies inflammation, increases the survival of plasmablasts capable of producing AQP4-IgG antibodies, supports the differentiation of B cells to plasma cells, and induces BBB injury [66,67]. Because B cells and autoantibodies were found to be disease beginners in experimental autoimmune encephalomyelitis (EAE), inebilizumab is instructive for modeling the therapeutic effects with enhanced ADCC against CD19-positive B cells, as confirmed in MS and NMOSD [68,69].

The activation of the complement cascade in patients with NMOSD was reported to increase membrane permeability and promote the influx of serum AQP4-IgG antibodies, which further amplified the inflammatory reaction at the BBB of the CNS [70]. Basic research demonstrated that AQP4 antibodies trigger the complement system and lead to MAC formation via the CDC pathway, which results in astrocyte damage and secondary neuronal injury [71,72]. C1q-targeted monoclonal antibodies were demonstrated to effectively inhibit AQP4-IgG-mediated CDC, which interfered with MAC, and also IgG-mediated CDCC, which influenced the formation of the Cb3–Cb3R complex on macrophage and NK cells in an in vivo study [73]. Eculizumab is a humanized monoclonal antibody that inhibits terminal C5 complement protein cleavage into the C5a (inducing proinflammatory activity) and C5b fragments (inducing the MAC formation) [74,75,76]. Serum C4 levels were found to be lower in patients with AQP4-IgG-positive NMOSD in clinical remission than in those with MOGAD and MS as well as in healthy controls [77]. Immune features and the cytokine profile in the cerebrospinal fluid (CSF) significantly vary in patients with MS, AQP4-positive NMOSD, and MOGAD, suggesting that these are different autoimmune demyelinating diseases [78,79]. The role of complement in MOGAD has not been fully elucidated so far. It is possible that MOG-IgG could cause reversible myelin damage without complement activation [80]. On the other hand, a subset of human MOG-IgG antibodies was shown to induce complement-dependent pathogenic effects in a murine animal model [81]. Increased levels of proteins indicating classic and alternative complement activation were observed in patients with MOGAD compared with the control groups. Therefore, complement activation could be a potential therapeutic target in patients with MOGAD [82].

The new evidence on NMOSD pathophysiology highlights promising treatment modalities as well as clinical studies [83]. Restoring immune tolerance might provide an interesting treatment strategy in the future [84]. Some success was achieved by using autologous hematopoietic stem cell transplantation [85], peptide-loaded tolerogenic dendritic cells [86], DNA vaccine encoding myelin basic protein, [87], autoreactive T cell vaccination, and regulatory T cells [88,89]. Further alternative targets for NMOSD treatments are the blood–brain barrier, [90], the complement cascade [91], and B cells [92].

## 4. Genetic Susceptibility to NMOSD

Despite important breakthroughs in the understanding of AQP4 and MOG antibodies and their involvement in NMOSD, the genetic factors underlying the disease pathogenesis have not been fully understood. More recently, genome-wide single-nucleotide polymorphism arrays have shown some susceptibility loci for NMOSD [93]. It is predominantly a sporadic disorder, although familial NMOSD occurs in 3% of the cases [93,94]. Human leukocyte antigen (HLA) haplotypes were reported to be highly correlated with NMOSD. HLA is located on chromosome 6, and the main variations are observed in DQA1, DQB1, DRB1, and DPB1 [95,96,97]. Whole-genome sequence studies that have been conducted in Europeans since 2009 identified a C4A deletion and a fourfold reduction of C4a levels as the most likely functional drivers of an increased risk for AQP4-IgG production. Furthermore, HLA-DQA1*102, HLA-DQA1*501, HLA-DQB1*0201, and HLA-DRB1*03 alleles were significantly associated with NMOSD [96,98]. In Japan, Ogawa et al. found that HLA-DQA1*05:03 was significantly associated with NMOSD, whereas Watanabe et al. reported that HLA-DRB1*08:02 and HLA-DPB1*05:01 were associated with susceptibility to NMOSD and that HLA-DRB1*09:01 was protective against NMOSD [99,100,101]. In addition, distinct genetic and infectious profiles in Japanese patients with NMOSD demonstrated that the HLA-DRB1*1602 and HLA-DPB1*0501 alleles as well as infection with *Helicobacter*
*pylori* and *Chlamydia pneumonia* were associated with higher susceptibility to AQP4-IgG-seropositive NMOSD [102]. Future studies should evaluate response to treatment as well as genetic and cytokine profiles in association with distinct genetic backgrounds in patients with NMOSD.

In summary, the potential molecular mechanisms underlying AQP4-seropositive NMOSD may be related to proteins encoded by the novel genes involved in complement activation, antigen presentation, antibody-dependent cytotoxicity, and immune regulation [103].

## 5. Potential Biomarkers in NMOSD

### 5.1. Surrogate Serum Biomarkers

Evidence shows that AQP-4 IgG antibodies are not strongly associated with clinical indices, such as the EDSS, risk of relapse, or visual prognosis in NMOSD [17]. On the other hand, there are data suggesting that the activation of complements, cytokines, and chemokines contributes to the complex pathogenesis of the disorder [104,105]. Naive T-helper cells differentiate into a new lineage called Th17 and have the capacity to produce large amounts of IL-17, a cytokine linked to autoimmune diseases [67,106]. IL-6 signaling involving Th17 cells and Th17-associated cytokines may play a crucial role in the pathogenesis of NMOSD (Figure 1) [66,107,108,109,110]. Apart from IL-17, Th17-cell differentiation may be induced by IL-6, IL-23, and transforming growth factor β1 [111,112]. Granulocyte macrophage colony-stimulating factor acts as a proinflammatory cytokine and could critically be involved in the formation of Th17 cells and the activation of macrophages and dendritic cells involving the secretion IL-23 and IL-6 [113]. Patients with NMOSD were shown to have higher levels of IL-6, IL-17, and IL-21 in both CSF and serum, as well as higher levels of IL-1, IL-8, IL-13, and granulocyte colony-stimulating factor in the CSF than those with MS [66,107]. The IL-6 levels in CSF correlate with neural damage biomarkers in NMOSD, and increased plasma IL-6 levels correlate with the EDSS [114]. As for the pathological mechanisms, IL6 signaling is thought to contribute in multiple ways, as shown in Figure 1. Two monoclonal antibodies, satralizumab and tocilizumab, activate the same mechanism, in that they both target IL-6 receptor- (IL-6R) and IL-6-associated immune cascades, leading to T-cell activation, IgG secretion, BBB damage, activation of the complement cascade, and enhancement of macrophage and microglia activity [84,109,115,116]. In a novel in vitro BBB model, the proposed role of IL-6 on the BBB was confirmed [117]. AQP4-IgG induced IL-6 release from astrocytes, then the BBB was impaired by IL-6 signaling in endothelial cells, and reversal of the BBB impairment was enhanced by anti-IL-6 receptor (IL-6R) antibodies [110,118,119].

Complex processes involving activated microglia ultimately promote the pathological course of NMOSD, and that suggests that microglia may serve as a therapeutic target in NMO [120]. Briefly, complement C3a secreted from activated astrocytes may induce the secretion of complement C1q and inflammatory cytokines by microglia, facilitating injury to microglia, astrocytes, oligodendrocytes, and neurons in an autocrine or paracrine manner [121].

Serum biomarkers including glial fibrillary acidic protein (GFAP) and neurofilament light chain (NfL) may help guide the design of effective therapies for the management of disease [22,26,122,123,124]. In a subgroup analysis, the CSF levels of IL-6, NfL, and GFAP were higher in AQP4-IgG-positive cases and might be used as indicators of disease activity, relapse risk, and therapy efficacy [124,125,126]. Factors involving the tight junctions seem to be other candidates for key biomarkers. Epidermal growth factor may be involved in the disruption of the BBB by downregulating claudin-5 in NMOSD, and women were shown to exhibit higher urinary levels of this factors, which might explain their greater susceptibility to NMOSD [127]. Interferon-γ reduces BBB integrity in cultured brain endothelial cells through Rho kinase-mediated cytoskeletal contraction, causing junction irregularity and cell–cell disconnections leading to deformity in adherence and tight junction proteins [128,129]. Serum vascular endothelial growth factor, myeloid progenitor inhibitory factor 1, and neuron–glia-related cell-adhesion molecule were positively associated with AQP4-IgG titers; thus, they could be potential biomarkers of NMOSD [127]. A study on plasma chemokine levels in NMOSD during remission confirmed that IL-1β and tumor necrosis factor α stimulate eosinophilic chemoattraction, suggesting that the elevated secretion of monocyte chemotactic protein (C–C motif chemokine ligand 13, CCL13) and eotaxins (CCL11 and CCL26) may be a critical step in eosinophil recruitment during remission [130].

### 5.2. OCT Biomarker

The international CROCTINO program uses OCT as a standardized method to assess the clinical course and pathophysiology of NMOSD as well as to monitor therapeutic efficacy [131,132]. OCT was shown to provide unique insights into the identification of foveal pitting in NMOSD likely due to the loss of Müller cells [133].

## 6. Optic Nerve Structure in NMOSD

The use of OCT to discriminate the microstructures of the retinal nerve fiber layer and GC-IPL has been debated in recent studies on NMOSD and MS. Moreover, while current therapies were demonstrated to improve the visual function after acute treatment, structural improvement remains an unmet need. GC-IPL thickness associated with visual ability in NMOSD-ON eyes leads to more severe retinal thinning and visual impairment than that found in MS [134]. A cross-sectional collaborative retrospective study reported that the average GC-IPL loss was 22.7 μm after the first ON attack, and the average loss after a recurrent episode was 3.5 μm, with noticeable subclinical GC-IPL loss in non-optic neuritis (NON) eyes [132]. NMOSD-NON eyes exhibited reduced thickness in the GC-IPL but not in the pRNFL compared with normal eyes, and relative changes in the parvocellular layer of NMOSD-NON eyes were not fully confirmed in recent studies [132,134,135]. Due to the presence of numerous confounding factors when determining the thickness of the anterior visual pathway, parafoveal segmentation on OCT may enable a more sensitive detection of neuronal loss and reflect a neurodegenerative reaction of retinal ganglion cells (RGC) and Müller cell damage in NMOSD [136,137]. Contrarily, pRNFL in the papillomacular bundle exhibited a reduction in MS-NON due to parvocellular axons being more vulnerable to energy depletion in MS studies [138,139,140]. GC-IPL damage on OCT spatial relationship and papillomacular bundle loss in neurodegenerative disease such as mitochondriopathy or neuroinflammmatory disease require further research to elucidate the damage in the megacellular and parvocellular layers in the lateral genicular nucleus [139,141]. There were no significant differences in the annual changes in mGC-IPL, pRNFL, total macular volume, and disc cup ratio in the NMOSD-ON eyes when comparing patients treated with different immunosuppressive therapies [142].

## 7. Macular Structure in NMOSD

The foveal structure on OCT has been discussed as a potential biomarker of NMOSD. The foveal structure, including foveal pit shape, depth or total macular volume (mm^3^), or central foveal thickness, may be an early diagnostic marker of NMOSD [136,143,144]. This hypothesis is supported by data from animal studies, which showed Müller cell death after intravitreal injection of AQP4 antibodies and revealed a lysosomal degradation mechanism for AQP4 loss on Müller cells [145].

In the parafoveal region, which is rich in astrocytes and AQP4-expressing Müller cells, a positive association between attack-independent neural loss and visual function was observed [57]. Microcystic macular degeneration (MMD) may contribute to loss in both high- and low-contrast visual testing after an NMOSD-ON episode [136,137]. The inner nuclear layer (INL) and outer retinal layer were thicker in the NMOSD-ON eyes compared with NMOSD-NON and control healthy eyes, due to the presence of MMD in the episode of ON [134,137,146]. MMD is predominantly localized in the INL, but also extends to the outer nuclear layer [136]. Interestingly, a previous study demonstrated that the INL thickness was negatively correlated with the GC-IPL content in NMOSD [147].

To date, there have been no cross-sectional and longitudinal studies to elucidate whether MMD is a temporary change or secondary to optic neuropathy. Although the pathophysiologic mechanism of MMD in NMO is complex, degeneration is believed to be caused by the disruption of the blood–retinal barrier and the transition of microglial cells to phagocytose apoptotic RGCs [148]. Another possibility is vitreomacular traction, but fluorescein angiography is needed to confirm the cause–effect relationship [149]. MMD exhibited no progression over a 20-month follow-up, and the risk of MMD was observed in 4.7% of patients with MS and in 13.3–26% of patients with NMOSD [136,137,150]. MMD is not a specific sign of NMOSD, and the exclusion of other secondary insults, such as uveitis, diabetes, or Fingolimod exposure, is required [150,151]. In hereditary optic neuropathy, MMD may be associated with vitreomacular traction or the epiphenomenon of optic atrophy, unrelated to inflammation or retrograde transsynaptic degeneration [152]. In advanced glaucomatous optic atrophy, MMD in the superior and nasal macular quadrants was also documented on OCT [147]. Furthermore, the INL cyst secondary to retinitis pigmentosa was not an uncommon sign [153,154].

To summarize, INL cystoid lesions are a nonspecific indicator of degenerative optic neuropathy or retinopathy, and MMD could be linked to Müller cell pathology in NMOSD [155,156,157]. However, further research is required to prove this hypothesis.

## 8. OCT Angiography in NMOSD

Subclinical vascular changes in the parafoveal retina might occur during an ON attack and could be associated with astrocyte damage with increased levels of sNfL/sGFAP [158]. Patients with NMOSD exhibited an enlargement of the foveal avascular zone independent of an ON attack [158]. This could be explained by damage of the blood–retinal barrier resulting from Müller cell loss, leading to the enlargement of the foveal avascular zone on OCT angiography in patients with NMOSD but not in those with MS [137,144,158,159]. A strong correlation between the deep vascular complex and visual acuity was reported, and decreased microvascular density in the superficial and deep vascular complex was significantly correlated with the frequency of NMO-ON attacks [160]. OCT angiography with a measurable analysis offers a new possibilities in the study of microvascular impairment in NMOSD and may become an objective clinical tool for patient monitoring.

## 9. Animal Models of Neuromyelitis Optica

The available animal models of NMO are largely based on a passive transfer of AQP4-IgG antibodies or AQP4-sensitized T cells to rodents and are often combined with proinflammatory maneuvers (coinfusion of proinflammatory factors or additional needle trauma) [60,161,162,163]. The models exhibit T cell and B cell activation, macrophage/microglial infiltration, eosinophil aggregation, immune complex deposition, loss of AQP4 and GFAP expression, and astrocyte/axonal injury [42,45,60,162] (Table 2). There are currently two main methods for generating animal models of NMO: one is NMO-IgG immunization in the EAE model [162,163,164,165,166], and the other is coinjection of the NMO-IgG/human complement into the target, which can be a route of intraventricular, intra-spinal cord, perichiasmal, or transoptic nerve sheath [42,92,161,167,168,169,170].

## 10. Animal Model of NMO

Passive immunization by intraperitoneal injection of AQP4-specific T cells in EAE rats allows EAE to develop faster via specific targeting of the astrocytes and the entry of pathogenic AQP4 antibodies to produce NMO-like lesions in the CNS after 10 to 14 days from the injection [171]. The extent and location of inflammation and demyelination mainly vary according to the specific antigen introduced and rodent species and strain [170]. Lewis rats can produce high titers of antibodies against specific epitopes of human AQP4 [172]. However, Saadoun et al. recently demonstrated that the coinjection of NMO-IgG with human complement could produce NMO-like lesions in naive mice without EAE [92]. Asavapanumas et al. reported that a single intracerebral or intraperitoneal injection of NMO-IgG antibodies after intracerebral needle stab injury without the administration of complement or proinflammatory cytokines was able to produce a mouse model of NMO [60,161]. Evidence suggests that the coadministration of the complement is required for developing a complete NMO lesion but is not needed for the development of the penumbra, which emphasizes the pathogenic role of macrophage/microglia involving ADCC in NMO [59,60].

Studies demonstrated that a passive transfer by intravenous or intraperitoneal injection of the AQP4 antibodies from seropositive NMO patients was insufficient to cause NMO-like lesions in rodents, as the low levels of antibodies could not penetrate into the CNS [163,173]. When applying patient sera without any filtration modification, it is likely that coexisting human complements may play a synergic role in the development of strong neuroinflammation [169,174,175]. Optic nerve susceptibility in NMO might also arise from the abundant AQP4 expression along the optic nerve compared with the brain [65]. Aside from the optic nerve, AQP4 is also highly expressed in astrocyte-like Müller cells in the inner retina and the ciliary epithelium [7]. A study using intravitreal AQP4-IgG passive transfer resulted in a complement-independent retinal pathology that reduced AQP4 expression and increased GFAP levels by 5 days [145]. We summarize the rodent models regarding ON without existence of EAE in Table 2. The passive cotransfer of NMO-IgG antibodies and complement via a continuous 3-day infusion near the optic chiasm in mice seemed to be sufficient to develop NMO-ON [42]. However, continuous infusion with precise needle placement is technically challenging and might cause additional irreversible damage. On the other hand, retrobulbar infusion, intravitreal injection, or a single intracranial injection may result in a limited or transient pathology [42]. We believe that local retrobulbar injection of NMO-IgG-positive serum is difficult to perform, as this approach easily causes optic nerve unpredictable traumas. However, Matsumoto et al. injected human NMO-IgG-positive serum directly into the space of the optic nerve sheath after desheathing, which led to infiltration of inflammatory cells into the optic nerve and regional astrocyte loss with progressive loss of RGCs and demyelination at day 14 [169].

## 11. Limitations of the Animal Models of NMO

Although each model was shown to have features of human NMO-like lesions, multiple factors may limit direct comparisons between animal and human data. First, in human NMOSD, AQP4-IgG antibodies are produced continuously, then astrocyte loss is associated with loss of myelin. Second, astrocytes and RGCs in the human CNS and eye are more complex than in rodents [176,177]. Third, the coinjection of AQP4-IgG antibodies and complement can activate the complement system in rats but not in mice and may lead to underestimation of the complement inhibitory function. Finally, EAE models in rats are Th1-cell-mediated, whereas AQP4-specific T cells in NMO reportedly show a preferential involvement of Th17/Th2 lymphocytes including IL-6- and Th17-polarizing cytokine interactions [65,108,178].

Animal models may enable us to understand the early pathogenetic mechanisms in the immune cascade of nerve inflammation and develop potential drug therapies; however, they partially recapitulate the pathological features of human NMO in animal model as the complement system and humoral/cytotoxic immunity vary between different species [170,172,179].

## 12. Conclusions

NMOSD is a complex multifactorial neuroinflammatory disease; extensive research is ongoing on the pathogenesis, genetic background, serum biomarkers, OCT segmentation. Novel drugs targeting the complement cascade system, IL-6R, and B cells are being studied. Restoring the blood–brain barrier and enhancing immune tolerance by using stem cell transplantation, dendritic cells, vaccine and regulatory T cells might provide potential therapeutic strategies in the future. Animal models may help gain a better understanding of the detailed immune mechanisms involved and could lead to the development of potential future treatments, for example, based on the inhibition of AQP4 antibody formation that could prevent the activation of inflammatory cells and offer neuroprotection.

## Figures and Tables

**Figure 1 ijms-23-07908-f001:**
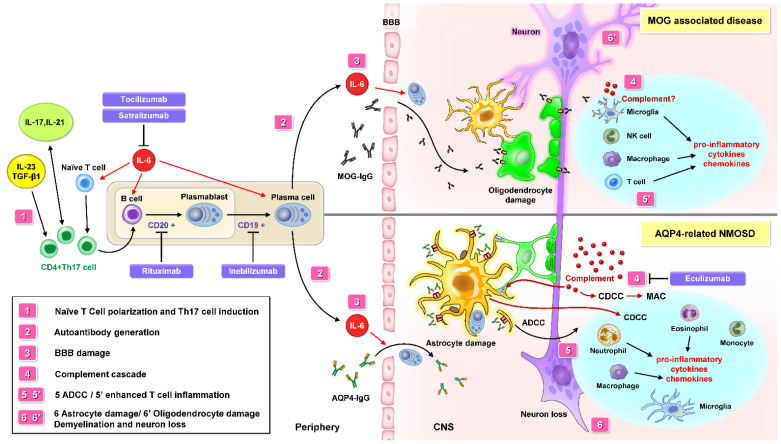
The pathological mechanism of NMOSD may involve peripheral autoimmune dysregulation. Interleukin 6 (IL-6) is a key factor in AQP4-related NMOSD pathophysiology. A similar role of IL-6 is also reported in MOG-associated disease (MOGAD). Besides IL-6, Th17 cells differentiation may be induced by IL-17, IL-21, IL-23, and TGF-β1. It is thought that an impaired innate immune system may promote naive T cell transformation into Th 17 cell and stimulate B cell differentiation to plasmablasts, then to plasma cells producing AQP4-IgG or MOG-IgG autoantibody. A leaky BBB contributes to the migration of AQP4-IgG from the periphery into the CNS. AQP4-IgG bind to AQP4 and activate the complement cascade (CDC and CDCC) and ADCC. Cytokine and chemokine production leads to the recruitment of macrophages, eosinophils, neutrophils, and monocytes to the inflammation site. After microglia and macrophage infiltration, astrocytes and oligodendrocyte are damaged, which leads to advanced axonal degeneration and neuronal death. AQP4-IgG-seropositivity in NMOSD indicates the entity of astrocytopathy, and MOG-IgG results in oligodendropathy, named MOGAD. Current maintenance therapies include interfering with complement activation (Eculizumab), IL-6 R signaling (Tocilizumab and Satralizumab), and plasma cells producing AQP-4 and MOG IgG Abs (Rituximab and Inebilizumab). Abbreviations: NMOSD = neuromyelitis optica spectrum disorders; BBB = blood–brain barrier; CDC = complement-dependent cytotoxicity; CDCC = complement-dependent cellular cytotoxicity; ADCC = antibody-dependent cellular cytotoxicity; MOG = myelin oligodendrocyte glycoprotein; AQP-4 = aquaporin-4; IL-6 = interleukin 6; IL-17 = interleukin 17; Th17 cell = T helper 17 cell; TGF-β1 = transforming growth factor beta 1; IL-6 R = IL-6 receptor.

**Table 1 ijms-23-07908-t001:** NMOSD diagnostic criteria for adult patients.

**Diagnostic criteria for NMOSD with AQP4 IgG** At least one core clinical characteristic Positive test for AQP-IgG using an available detection method (CBA recommended) Exclusion of alternative diagnoses
**Diagnostic criteria for NMOSD without AQP4-IgG or NMOSD with unknown AQP4-IgG status** At least two core clinical characteristics occurring as a result of one or more clinical attacks and meeting all the following requirements: At least one core clinical characteristic must be optic neuritis, acute myelitis with longitudinal extensive neuritis, acute myelitis with LETM, or area postrema syndrome Dissemination in space (two or more different core clinical characteristics) Fulfillment of additional MRI criteria * Negative tests of AQP4-IgG using an available detection method, or testing unavailable Exclusion of alternative diagnoses
**Core clinical characteristics** Optic neuritis Acute myelitis Area postrema syndrome: episode of otherwise unexplained hiccups or nausea and vomiting Acute brainstem syndrome Symptomatic narcolepsy or acute diencephalic clinical syndrome with NMOSD-typical diencephalic MRI lesions Symptomatic cerebral syndrome with NMOSD-typical brain lesions
Modified IPND 2015 NMOSD Criteria [13]. *** Additional MRI criteria** Acute optic neuritis: requires brain MRI showing normal findings or only nonspecific white matter lesions, or optic nerve MRI with T2-hyperintense lesion or T1-weighted gadolinium-enhancing lesion extending >1/2 optic nerve length or involving optic chiasm. Acute myelitis: requires associated intramedullary MRI lesion extending ≥3 contiguous segments (LETM) OR ≥3 contiguous segments of focal spinal cord atrophy. Area postrema syndrome: requires associated dorsal medulla/area postrema lesions. Acute brainstem syndrome: requires associated periependymal brainstem lesions.

Abbreviations: NMOSD = neuromyelitis optica spectrum disorders; AQP4 = aquaporin-4; LETM = longitudinal extensive transverse myelitis; CBA = cell-based assay.

**Table 2 ijms-23-07908-t002:** Animal models of NMO-optic neuritis without experimental autoimmune encephalomyelitis.

Reference	Animal	Model System	Significance
Matsumoto et al., 2014 [169]	Adult Lewis rats	NMO patients’ sera were applied on the optic nerve after desheathing	7 days after treatment: lost expression of both AQP4 and GFAP on IHC, leading to regional astrocytic degeneration and inflammatory cell invasion, which resulted in secondary loss of RGCs and their axons
Asavapanumas et al., 2014 [42]	8- to 10-week-old, weight-matched AQP4+/+ and AQP4−/− mice in CD1 genetic background	Passive transfer of NMO-IgG and complement by continuous 3-day intracranial infusion near the optic chiasm	Loss of AQP4 and GFAP immunoreactivity, granulocyte and macrophage infiltration, deposition of activated complement, and demyelination and axonal loss
Asavapanumas et al., 2014 [161]	Adult Lewis rats	A single intracerebral needle insertion, without pre-existing inflammation or infusion of proinflammatory factors	At 5 days, there was marked loss of AQP4, GFAP, and myelin. Granulocyte and macrophage infiltration, complement deposition, BBB disruption, microglial activation, and neuron death. The penumbra was associated with a complement-independent mechanism (antibody-dependent cellular cytotoxicity).
Saadoun et al., 2010 [92]	8- to 10-week-old, wild-type and AQP4-null mice on a CD1 genetic background	Intracerebral coinjection of Ig G from NMO patients with human complement	Within 12 h of injection, striking loss of AQP4, glial cell edema, demyelination, and axonal loss, but little intraparenchymal inflammation. At 7 days, there was extensive inflammatory cell infiltration, perivascular deposition of activated complement, extensive demyelination and loss of astrocytes, and neuronal cell death.

Abbreviations: NMO = neuromyelitis optica; AQP4 = aquaporin-4; IHC = immunohistochemistry; RGCs = retinal ganglion cells; GFAP = glial fibrillary acidic protein; BBB = blood–brain barrier.

## Data Availability

Not applicable.

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
