# Peer review of "Neuromyelitis Optica Spectrum Disorder: From Basic Research to Clinical Perspectives"

_ijms, 2022, doi:10.3390/ijms23147908_

Round 1

Reviewer 1 Report

You present a comprehensive review on a topic of great international interest y actuality. Your paper it is well-designed and the flow of the material is appropriate. There are some segments of the manuscript where the text should be more clear and informative for the reader. Even though the reader may obtain the missing information by consulting the references provided, the authors should provide the missing information as part of the general introduction. (1) Page 3, 2nd paragraph: the extent of the optic nerve involvement in positive anti-AQP4-IgG and anti-MOG-IgG is not discussed. In the same paragraph I would advise more extensive discussion between the sex ratio distributions among these disorders. (2) Explain what are the different anatomical distributions of involvement in children and adults referring to the maturation location of M1 and M23 proteins (page 4 first paragraph). This is mentioned but no discussed. (3) Please review your references for clarity. For instance reference # 4 is written defectively. 

Reviewer 2 Report

authors mentioned satralizumab and tocilizumab, it would be interesting to discuss their mechanism of action

in conclusion section, I would suggest to develop perspectives of such therapies

Could you develop the role of astrocyte in the disease, also it would be interesting to discuss how micrgolia/immune cells affect the astrocytes and the disease
